# Improved Design of Electroforming Equipment for the Manufacture of Sinker Electrical Discharge Machining Electrodes with Microtextured Surfaces

**DOI:** 10.3390/ma18091972

**Published:** 2025-04-26

**Authors:** Mariana Hernández-Pérez, Pedro M. Hernández-Castellano, Jorge Salguero-Gómez, Carlos J. Sánchez-Morales

**Affiliations:** 1Integrated and Advanced Manufacturing Research Group, University of Las Palmas de Gran Canaria, Canary Island, 35001 Las Palmas, Spain; pedro.hernandez@ulgpc.es (P.M.H.-C.); carlos.sanchez@ulpgc.es (C.J.S.-M.); 2Materials and Manufacturing Engineering and Technology Research Group, University of Cádiz, 11519 Puerto Real, Spain; jorge.salguero@uca.es

**Keywords:** micro-electroforming, microtexturing surfaces, SEDM, stereolithography by mask (MSLA)

## Abstract

The development of microtextures has had a transformative impact on surface design in engineering, leading to substantial advancements in the performance, efficiency, and functionality of components and tools. This study presents an innovative methodology for fabricating SEDM electrodes. The methodology combines additive manufacturing by mask stereolithography with an optimized electroforming process to obtain high-precision copper shells. A key aspect of the study involved redesigning the electroforming equipment, enabling the independent examination of critical variables such as anode–cathode distance and electrolyte recirculation. This approach allowed precise analysis of their impact on metal deposition. This redesign enabled the assessment of the impact of electrolyte recirculation on the quality of the shells obtained. The findings indicate that continuous recirculation at 60% power effectively reduced thickness deviation by up to 32.5% compared to the worst-case scenario, achieving average thicknesses within the functional zone of approximately 110 µm. In contrast, the absence of flow or excessive turbulence did not generate defects such as unfilled zones or non-uniform thicknesses. The shells obtained were validated as functional tools in SEDM, demonstrating their viability for the generation of textures with high geometric fidelity. This approach optimizes the manufacturing of textured electrodes and opens new opportunities for their application in advanced industrial processes, providing a more efficient and sustainable alternative to conventional methods.

## 1. Introduction

The optimization of surfaces through functional texturing has become a priority objective in engineering due to its direct impact on the efficiency, lifetime, and performance of industrial components. These improvements are essential in sectors such as the aerospace and automotive industries, where the efficiency and durability of parts directly influence performance and operating costs [1].

The design of textured surfaces has been significantly influenced by natural phenomena, such as the superhydrophobicity observed in lotus leaves or the adhesion mechanisms of certain reptiles. These natural microtextures have been studied to optimize various surface properties [2,3]. These biomimetic strategies have been implemented in the development of geometric patterns that allow, for example, the formation of lubricating microdeposits and the reduction in stress concentrations in the contact zone, thus minimizing wear and improving the operating performance of the systems [4].

Studies have shown that modifying surface topography by texturing techniques can significantly reduce friction and improve wear resistance by promoting homogeneous load distribution and thermal dissipation [5]. Depending on the material and the desired effect, there are several methods for surface texturing, such as subtractive and additive methods. However, the fabrication of these textures faces technical, economic, and environmental challenges that demand innovative solutions.

Among the subtractive processes, traditional techniques such as milling, laser engraving, and EDM stand out because they allow the generation of precise patterns in a wide range of materials [6,7]. However, they have significant limitations. Mechanical milling, for instance, leads to high operating costs due to tool wear and the time required for complex geometries [8]. Lasers, while offering versatility, require substantial energy and generate thermal waste that can compromise the integrity of the material, leading to increased surface roughness and potential performance issues [9]. Sinking EDM (SEDM) involves a significant consumption of graphite or tungsten electrodes, which are costly and challenging to recycle [10]. These processes often generate metallic waste and polluting emissions, which can compromise long-term sustainability [11]. In light of these challenges, there is a growing interest in developing more economical and sustainable processes that enable the precise fabrication of textured surfaces. In this regard, additive manufacturing has emerged as a promising technology. One such example is mask stereolithography (MSLA) technology, which allows the creation of models with high resolution and geometric complexity, facilitating the design of micro- and nanostructured surfaces [12]. Integrating these methods with electrochemical processes, such as electroforming, has opened new perspectives in the fabrication of electrodes and molds with functional texturing [13,14].

Electroforming is a method for producing free-form solid objects by means of electrolytic deposition on a cathode. This process forms a thin piece of material that replicates the geometry of the original substrate with high precision [15]. This metallic layer is then removed, resulting in the coating itself becoming the final product.

The use of copper in the fabrication of electrodes for electroforming processes offers significant advantages in terms of sustainability and performance compared to materials such as the aforementioned graphite used for SEDM. Copper’s exceptional electrical and thermal conductivity facilitates more efficient energy transfer during the SEDM process. Additionally, its high density and mechanical strength contribute to reduced electrode wear, prolonging the lifespan of the electrodes and decreasing the frequency of replacements [16].

The SEDM technique and its microscale variant have proven to be key tools for the fabrication of complex geometries without direct mechanical contact, which reduces stress generation and consequent material deformation [17].

Additive manufacturing has demonstrated significant potential in the production of structures with complex geometries and high fidelity, which is essential for the fabrication of tools with special requirements, including SEDM electrodes [18]. However, scientific literature lacks information regarding the production of textured SEDM tools by MSLA and electroforming, particularly concerning critical parameters such as the anode–cathode distance and the impact of recirculation on the quality and homogeneity of the deposition. The absence of systematic studies that address the interaction between these factors in laboratory equipment specifically designed for this purpose represents a significant opportunity for advancing the technique. In this context, the present study details the redesign of laboratory equipment, which allowed the performance of several analyses, such as the effect of recirculation on obtaining copper shells suitable for application as tools in SEDM. The study also presents preliminary findings from using one of these shells as a finishing tool; this has contributed to the identification of key parameters for optimizing this technique.

## 2. Materials and Methods

The present methodology is structured in several phases, as illustrated in Figure 1. These phases range from the generation of the initial texturing to the inspection of the eroded part, passing through intermediate phases, such as the electroforming process, which is the central stage of the study. The overall success of the procedure depends on the ability to produce a functional textured tool in both the preliminary and final phases.

The strategy for obtaining the electrodes is based on methodologies used in previous studies (see Sánchez et al. [19]). The different phases of the process are described briefly below.

### 2.1. Design and Fabrication of Functional Models

The SEDM electrode fabrication process commences with the design of model parts incorporating textured surfaces, which are adjusted to the functional requirements and the capabilities of the processes involved. For this purpose, computer-aided design tools (Fusion360 v.2.0.19941) were used to allow the direct incorporation of micrometer-scale textures on the functional surface. This approach ensured precise reproduction of the desired patterns. As illustrated in Figure 2, various microtextures were designed and grouped into three main categories: aesthetic, geometric, and biomimetic patterns. During the course of the study, some of these textures were optimized through adjustments to critical parameters, such as maximum depth, repetition frequency, and pattern orientation, to improve the replicability and functionality of the final parts.

The model part is then fabricated using the stereolithography technique, which establishes the appropriate operating parameters for the equipment and the resin. The equipment used (Phrozen Mini 8K, Phrozen Technology Co. Ltd., Hsinchu, Taiwan) offered a resolution of 22 microns per pixel and layer heights of up to 10 microns, enhancing the level of detail in the geometric patterns. Following this, a rigorous post-processing protocol was implemented, involving a centrifugal alcohol cleaning step, a 10 min ultrasonic cleaning step, a compressed air drying step, and finally a curing step in a dedicated station. The equipment utilized in this process is illustrated in Figure 3.

### 2.2. Sputtering

After obtaining the resin model part, the functional surfaces were metallized to enable the flow of electric current that favors the metallic deposition during the electroforming process. This step was carried out using the sputtering process, which deposits an ultra-thin layer, of the order of a few nanometers, on the active areas without altering the geometry of the textures. The sputtering coating equipment SC7620 ‘Mini’ from Quorum Technologies (Laughton, UK) was used for this purpose, applying a layer of gold–palladium alloy with a minimum thickness. To ensure homogeneous coverage, two depositions of 120 s each were carried out, using a current intensity of 18 mA, which ensured complete metallization of the textured surface (see Figure 4).

### 2.3. Electroforming

Once plating is complete, the model part is ready to be immersed in the electrolytic bath for the initiation of the electroforming process, which follows the steps illustrated in Figure 5.

Based on prior research, a proprietary equipment design was developed and constructed, as illustrated in Figure 6. The system’s fundamental components include a power supply, a specific support for the model parts, a mechanism for stirring the electrolytic bath, an automated device for controlling the stirring (DCAB), and a monitoring system that continuously records the intensity and voltage parameters.

The operation of the process is initiated by executing a sequential program that establishes various deposition stages. During the process, the power supply monitors the electrical parameters. At the end of the total programmed time, the process is interrupted and the model part, now coated with a copper shell, is removed.

However, during the experimental development, critical variables were identified that merit individualized study. These include the effect of electrolyte recirculation and the distance between the anode and the cathode. These parameters directly affect the thickness and uniformity of the copper deposit. To address these issues, the starting equipment was redesigned. This redesign involved implementing a two-compartment electrolyte division and incorporating a particle filter. This allows regular and scheduled bath renewals and bath renovation to avoid stratification effects (see Figure 7).

The smaller upper tank, equipped with a nozzle for evacuation, holds one liter. The electrolyte is continuously recirculated by a pump to the filter. Once purified, the bath returns to the upper tank through the orifices integrated in the agitation system. This configuration allows the electrolyte quality to be maintained for extended periods of time, favoring more uniform deposition. Regarding operating parameters, the pump commences operation at 50% power and continues up to 100%. For the different studies, three power levels were used: moderate (60% with a flow of 12.6 L/h), medium (70% with a flow of 13.6 L/h), and high (90% with a flow of 15.0 L/h).

The redesign was extended to the agitation system and the cathode clamping device, which were developed using MSLA technology. Figure 7 illustrates not only the new equipment but also the location and design of these components in the new equipment configuration. In addition, a specialized support has been engineered to facilitate the attachment of both the cathode and the agitation mechanism in a versatile manner, which is essential for the planned tests (see Figure 8).

Conversely, an intelligent switch was incorporated into the DCAB, thereby optimizing the automation of the process. This programmable device allows users to define schedules for activating and deactivating the pump switch, enabling tests to be conducted without operator supervision. Therefore, two recirculation regimes were studied: continuous, with the pump running continuously, and intermittent, where a cycle of 10 min of active flow followed by 5 min of rest was used. While these parameters are adapted to the experimental system and the electrolyte used, the approach is adaptable to other systems by means of proportional adjustment of the volume, viscosity of the electrolyte, and tank geometry.

The new system incorporates a planning of electroforming stages, in which a constant amperage is established and progressively increased during the process, culminating in a final intensity of 1 A (see Table 1). The initial stage is extended to ensure the formation of a homogeneous base layer, which serves as a foundation for subsequent deposits, thus guaranteeing complete coverage, especially in areas with low-relief textures that present greater challenges in filling the coating.

### 2.4. Use of the Shells as Texturized Tools for SEDM

Once the shells have been obtained with the appropriate thickness for conversion into tools, they are conditioned according to a specific protocol that guarantees their structural integrity and functionality. Prior to separation of the shell from the model part, the back of the shell is reinforced by applying an epoxy putty. This reinforcement is crucial to prevent deformation during separation and to facilitate subsequent assembly in the tool holder designed for EDM. Figure 9 illustrates each stage of the process in detail.

### 2.5. Validation Method

To validate the electroformed shells, a geometrical analysis of the textures, mass, and thickness distribution was performed.

Mass determination was carried out using an analytical balance (Mettler-Toledo AB204-S, Mettler-Toledo LLC, Columbus, OH, USA), which provided accurate and reliable measurements.

To evaluate the fidelity in the replication of textural details, a microscopic inspection of the shells was performed. Initially, an electronic measuring microscope (Olympus BX51, Olympus Corporation, Tokyo, Japan) with a magnification of 20× was used, which made it possible to obtain a preliminary view of the main textured surfaces to be studied (see Figure 10).

Subsequently, an exhaustive analysis was carried out using the high-precision Alicona InfiniteFocusG5+ variable-focus 3D optical microscope (Berlin, Germany), an instrument capable of measuring geometric features at micrometer and submicrometer scales. It has a vertical resolution of up to 10 nm and a capacity to measure roughness as small as 0.03 µm [20]. This equipment enabled the acquisition of point clouds, which, when analyzed with the MountainsMap v7.4 software (Digital Surf, France), facilitated the quantification of the microtopography of textured surfaces, as illustrated in Figure 11.

To determine the thickness of the shells, measurements were taken at multiple points using a mechanical comparator (Mitutoyo ID-C112B, Mitutoyo Corporation, Kawasaki, Kanagawa Prefecture, Japan) mounted on a calibrated test bench. This equipment, accompanied by a calibration certificate Nº 862, issued by the Metrology and Calibration Service of the University of Las Palmas de Gran Canaria, ensured the accuracy of the measurements. Figure 12 shows the entire measuring system in service conditions.

## 3. Results

The implementation of the new electroforming equipment enables more precise control of the variables involved in the process, allowing specific studies to focus on parameters of interest and discard those that are not relevant.

The double tank design exhibited enhanced stability during tests, as the dividing lid between the two containers acted as a barrier, protecting the electrolyte medium from the environment. This effect was reinforced by the built-in filter, which prevented the accumulation of particles and, consequently, avoided possible distortions in the experimental results. The results of a study on the effect of the intensity of recirculation on the quality of the copper shells obtained are presented below.

### 3.1. Analysis of the Impact of Pump Power on Recirculation

In complementary studies, the influence of the distance between the anode and the cathode was studied with the aim of determining the distance that optimizes the obtaining of coatings with greater thickness and homogeneous distribution. The results showed that, for this particular type of equipment, a distance of about 60 mm was the most favorable, since a significant increase in the mass of the coating was observed (see Figure 13).

With the optimum distance between the anode and cathode set at 60 mm, the impact of electrolytic bath recirculation on the thickness distribution in the obtained shells was analyzed. To this end, a range of power levels (60%, 75%, and 90%) were evaluated under both continuous and intermittent modes. Additionally, a control group without recirculation was included for reference, as outlined in Table 2.

Each experiment was performed three times, resulting in 21 trials. The model part used was the incoming hemisphere texture, as shown in Figure 4c. The resulting copper shells were very homogeneous, as they all averaged 3.2 g in mass. However, although the mass appeared to be the same, there were significant differences in the uniformity of the thickness of the different shells. Systematic measurements were made at 64 points per shell (eight columns by eight rows), and the results were organized into heat maps (see Figure 14).

Figure 15, Figure 16, Figure 17 and Figure 18 illustrate the results of seven of the tests performed in one of the experimental groups, with color increasing with increasing shell thickness.

The data demonstrate that, under 60% power and continuous recirculation, optimal distribution is achieved in terms of uniformity and functionality. In contrast, the intermittent mode at the same power resulted in more pronounced spatial heterogeneity, particularly with the appearance of the edge effect.

As the power was increased to 75%, a general trend of increasing thickness was observed, although accompanied by greater variability, particularly under the intermittent regime, where peripheral regions gained thickness at the expense of a significant loss in the center of the shell. This pattern was particularly pronounced at 90% power, with central zones exhibiting thicknesses below 70 µm at certain points. These conditions were associated with a partial loss of refilling in low-relief areas; so, this power regime was considered the upper operational limit of our equipment.

In the test conducted without recirculation, although a higher average thickness was achieved, the distribution was irregular, and multiple defects were observed in the center of the part, which is consistent with incomplete filling. Examples of these deficiencies are shown in Figure 19.

As Figure 20 illustrates, the average thickness deviation for each test indicates that the continuous recirculation regime at 60% power exhibited optimal performance.

### 3.2. Electrode Wear Study

On the basis of the tests carried out, another concrete with a shark’s skin texture was produced on a relief, using continuous recirculation at minimum power, and the dimensional variations of the shell were studied after the application of the EDM process, with the aim of quantifying the wear induced.

From this initial test, it could be observed that the variations suffered in the microtexture dimensions were of the order of a few microns, since the EDM process was carried out in finishing conditions, using very low VDI values to minimize tool deterioration (see Figure 21).

Only the first stage of erosion was analyzed, a stage in which the surface irregularities remaining from the manufacture of these tools tend to diminish. This phenomenon is explained by the initial concentration of electrical discharges on the protrusions, which induces the rounding of the sharp edges. Once the electrode surface is smoothed, the current distribution becomes more uniform in subsequent operations, progressively reducing wear.

## 4. Discussion

The results obtained in this study are consistent with previous research that has investigated the influence of electrode spacing on electroforming processes. For example, Permadi et al. found that a spacing of 50 mm is optimal to obtain coatings with high adhesion and uniformity [21]. Similarly, Ortega’s work points out that the combination of electrode spacing and orientation directly influences the homogeneity of the deposit, with an improvement in uniformity in the central region when the separation between the anode and cathode is reduced [22]. In addition, Yang et al. [23] showed that adjusting the distance between these electrodes, as well as their surface areas, can significantly improve the uniformity of the deposit thickness. This adjustment is particularly important in applications where high geometric accuracy is required over large areas. These results highlight the importance of finding a balance between process efficiency and coating quality by adapting the experimental setup to the material properties and system conditions [24]. In addition, the shortest distance of the developed equipment gave the best results; this result is in agreement with the conclusions of the aforementioned authors.

In addition, the influence of the electrolyte flow on the electroforming process has been documented in the literature. Sun and Wang, through theoretical models, demonstrated that the flow field modifies the ionic concentration distribution and the morphology of the deposit, an aspect that is particularly relevant for microstructures with different aspect ratios [25]. Likewise, McGeough and Rasmussen [26] analyzed the optimal conditions for obtaining uniform thickness coatings on complex surfaces, highlighting the importance of continuous electrolyte recirculation to preserve the chemical composition and reduce defects during deposition. This is in agreement with the results obtained in the tests, where the coatings obtained with continuous recirculation and low power showed a lower standard deviation and therefore a higher thickness homogeneity.

On the other hand, computational studies carried out by Chalupa et al. suggest that a precise control of the geometry of the recirculation nozzles and the optimization of the exit angles favor the achievement of a laminar flow, which contributes to improving the homogeneity of the coating and minimizing irregularities in the deposition process [27]. This is in agreement with the redesigns carried out in the electroforming equipment, since a high intensity of recirculation generates a turbulent flow that is unfavorable for metal deposition.

Finally, according to the conclusions of Yarlagadda et al., the copper electrodes obtained by electroforming have suitable characteristics for their application in EDM processes. However, their use in roughing operations is not recommended due to the limitations inherent in their morphology and mechanical properties [10].

## 5. Conclusions

This study demonstrates that the integration of low-cost additive manufacturing techniques with copper electroforming processes enables the efficient and economical production of microtextured electrodes for SEDM. These tools are fully functional and capable of producing micron-level surface textures in complex geometries that would be difficult to achieve with traditional machining technologies. In addition, the combination of these technologies allows production times to be reduced compared to conventional methods.

By modifying the experimental equipment, it was possible to isolate and control several critical process variables. Tests carried out with different electrolyte recirculation strategies have shown that controlled flow is essential to achieve homogeneity in the thickness of the deposited layer. In particular, it was found that intermittent recirculation with moderate pumping power (60%) improves material distribution and avoids defects such as edge stratification and the formation of zones of lower deposition. Conversely, excessive recirculation (90%) in both recirculation regimes has been observed to generate turbulence, thereby compromising the uniformity of the deposit. This underscores the critical importance of the meticulous regulation of this parameter to ensure process stability.

A preliminary analysis of the initial wear of the electrodes after a spark erosion test showed the need for more detailed studies, including a larger number of tests to evaluate the wear progression in successive spark erosion processes under different operating conditions. This will make it possible to determine the service life of this type of electrode according to the required accuracy of the textures, as well as the required minimum thickness of the electroformed case according to the specific application and the expected service life of the electrodes.

## Figures and Tables

**Figure 1 materials-18-01972-f001:**
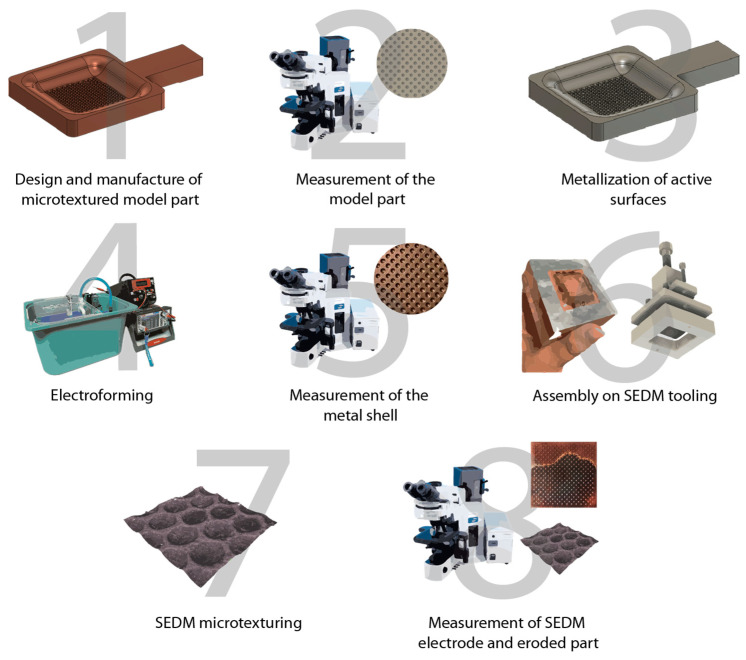
Flow required for the production of electroformed SEDM electrodes.

**Figure 2 materials-18-01972-f002:**
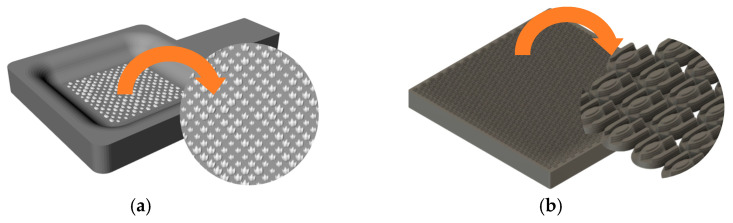
Example of microtextures: (**a**) parameterized sharkskin texture; (**b**) optimization of sharkskin texture by reducing its size and increasing the matrix.

**Figure 3 materials-18-01972-f003:**
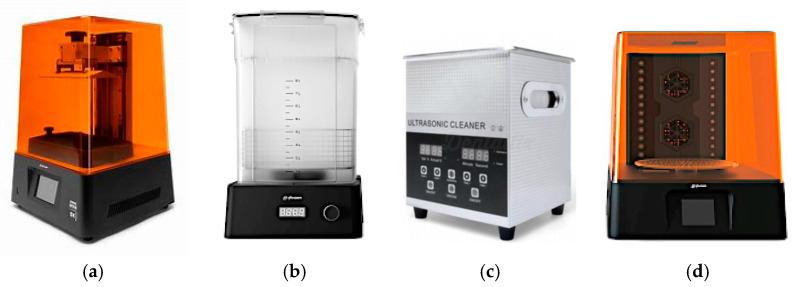
Equipment used to manufacture the model part: (**a**) Phrozen Mini 8K; (**b**) Phrozen Washing Station; (**c**) Phrozen Ultra-Sonic Cleaner; (**d**) Phrozen Dry and Cure Kit.

**Figure 4 materials-18-01972-f004:**
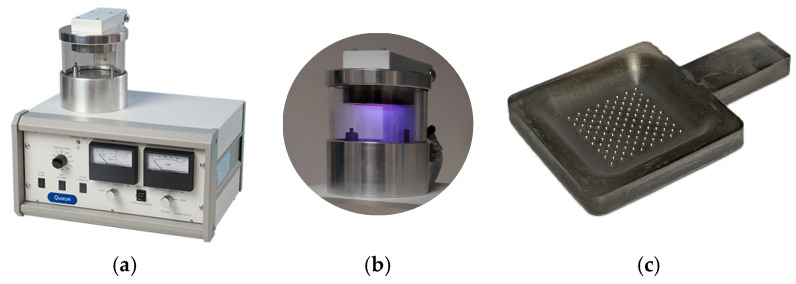
Sputtering: (**a**) SC7620 Mini Sputter Coater; (**b**) sputtering process; (**c**) metallized model part.

**Figure 5 materials-18-01972-f005:**
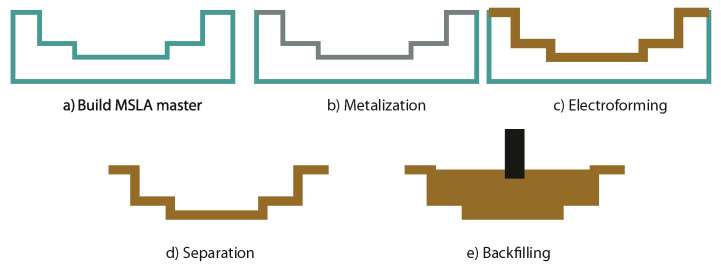
Electroforming process for SEDM tooling generation.

**Figure 6 materials-18-01972-f006:**
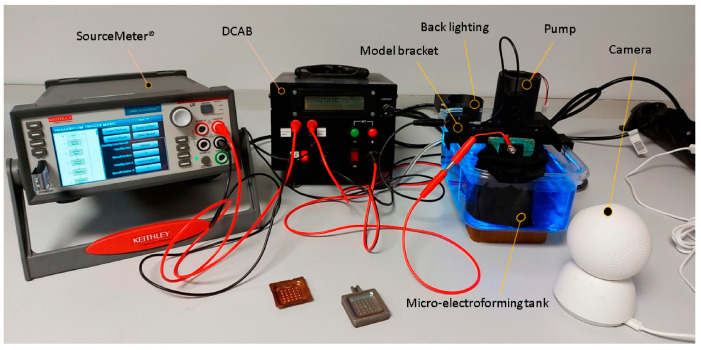
Initial electroforming equipment. Image obtained from [19].

**Figure 7 materials-18-01972-f007:**
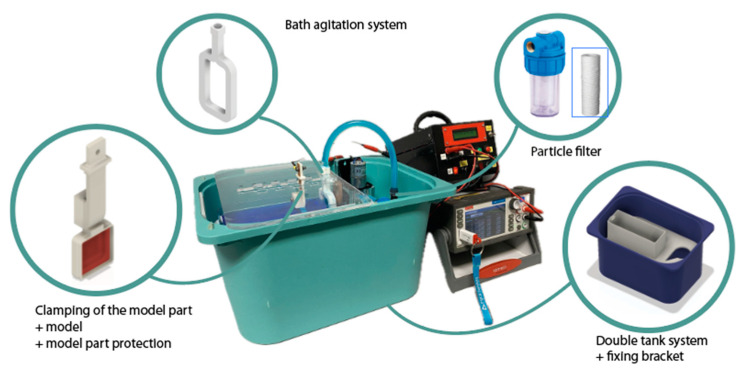
Redesigned electroforming equipment.

**Figure 8 materials-18-01972-f008:**
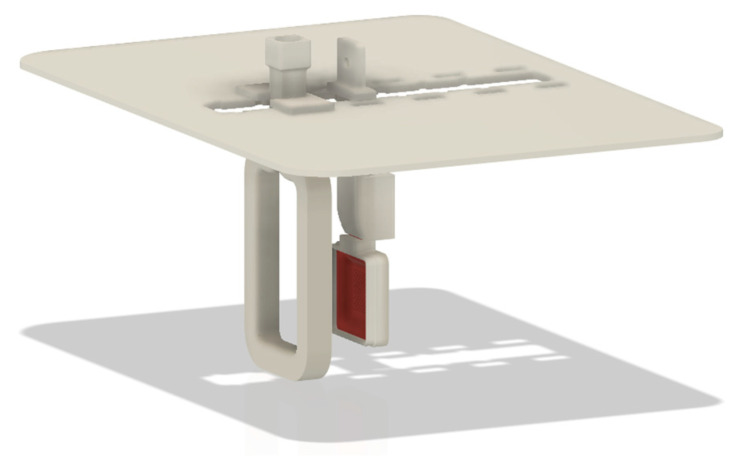
Support for cathode placement and bath agitation element.

**Figure 9 materials-18-01972-f009:**
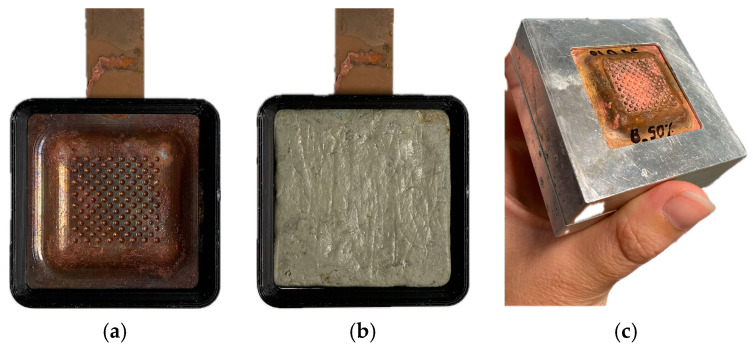
Copper shell reinforcement: (**a**) removal of the copper-plated model part; (**b**) filling with epoxy putty; (**c**) insertion of the shell into the tool holder.

**Figure 10 materials-18-01972-f010:**
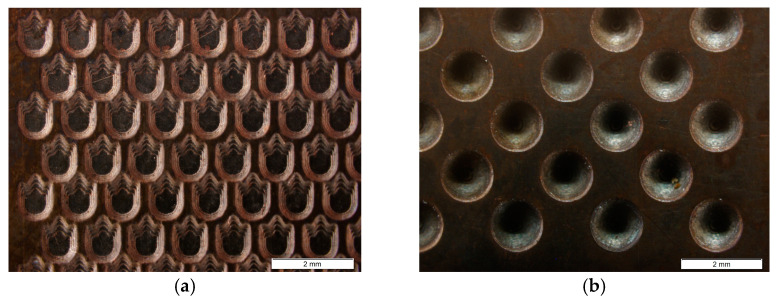
Visual inspection of shells with different microtexture designs: (**a**) texture on shark skin relief; (**b**) texture on hemisphere relief.

**Figure 11 materials-18-01972-f011:**
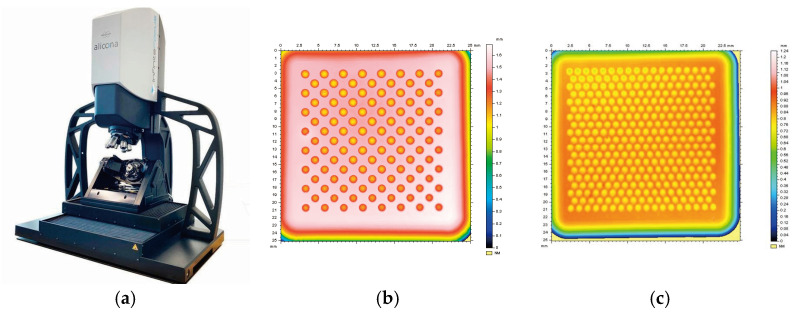
Visual inspection of shells by microscopy through MountainsMap: (**a**) Alicona InfiniteFocusG5+ equipment; (**b**) texture under relief; (**c**) texture over relief.

**Figure 12 materials-18-01972-f012:**
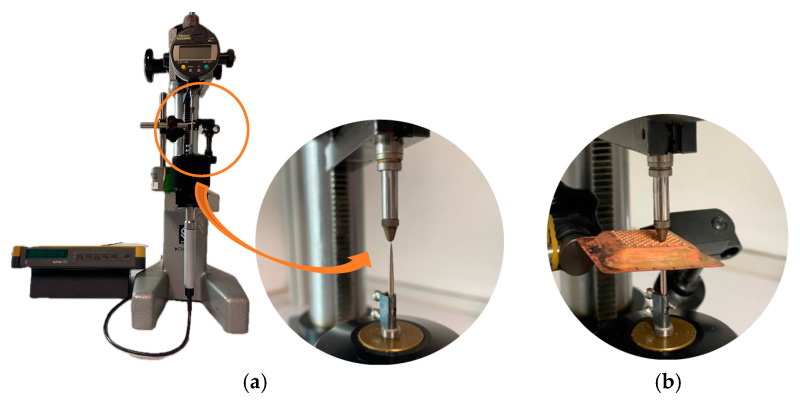
Thickness measurement of copper shells: (**a**) enlargement of the stylus used for measurement; (**b**) example of shell measurement.

**Figure 13 materials-18-01972-f013:**
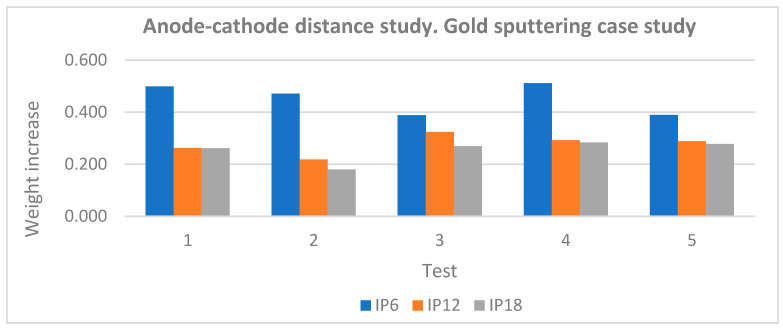
Anode–cathode distance study, where IP refers to the weight increase for the different distances of 60, 120, and 180 mm.

**Figure 14 materials-18-01972-f014:**
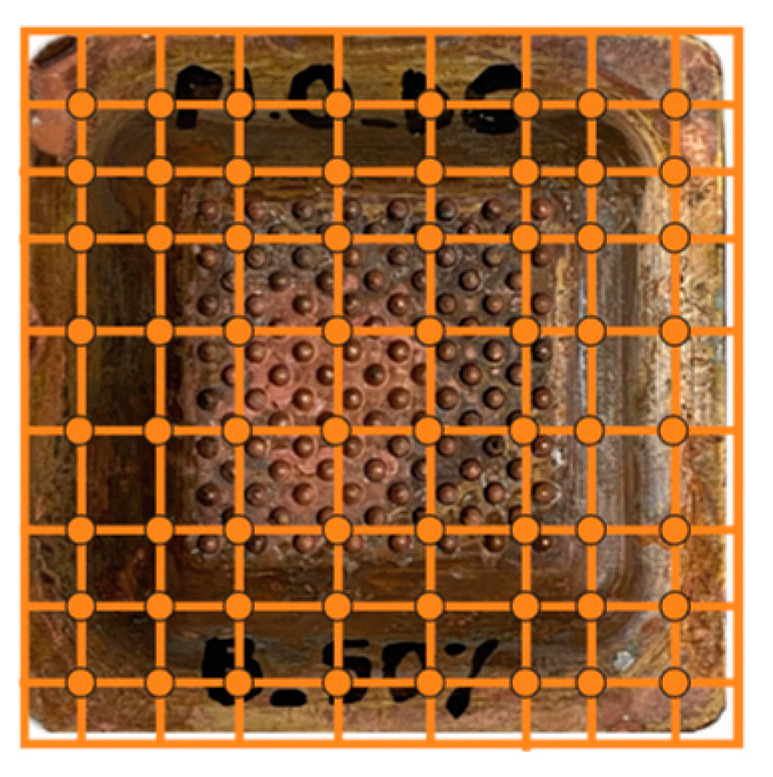
Distribution of points for taking measurements of electroformed shells.

**Figure 15 materials-18-01972-f015:**
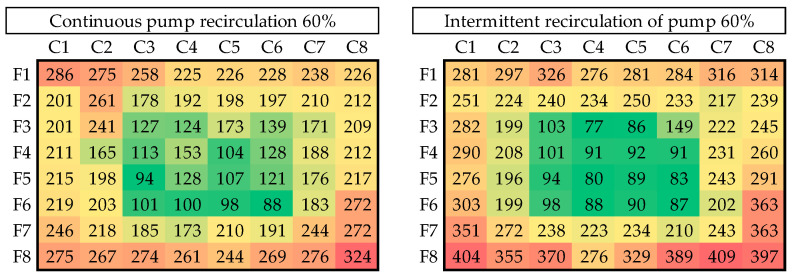
Thickness distribution for continuous and intermittent recirculation at 60% power.

**Figure 16 materials-18-01972-f016:**
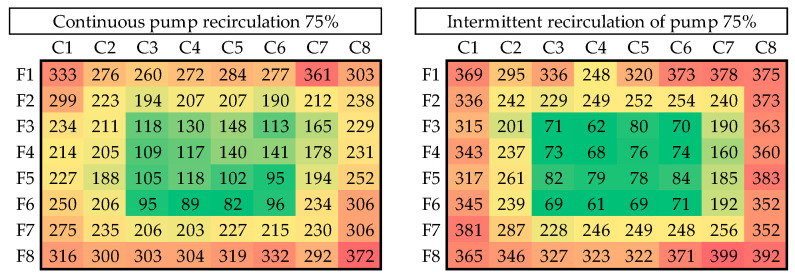
Thickness distribution for continuous and intermittent recirculation at 75% power.

**Figure 17 materials-18-01972-f017:**
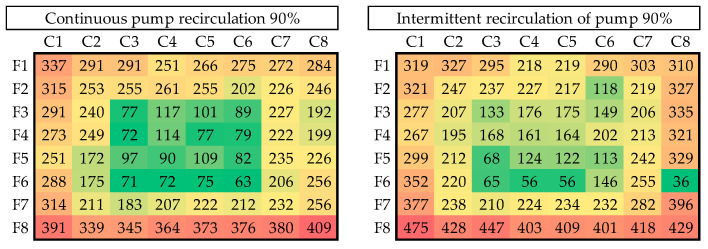
Thickness distribution for continuous and intermittent recirculation at 90% power.

**Figure 18 materials-18-01972-f018:**
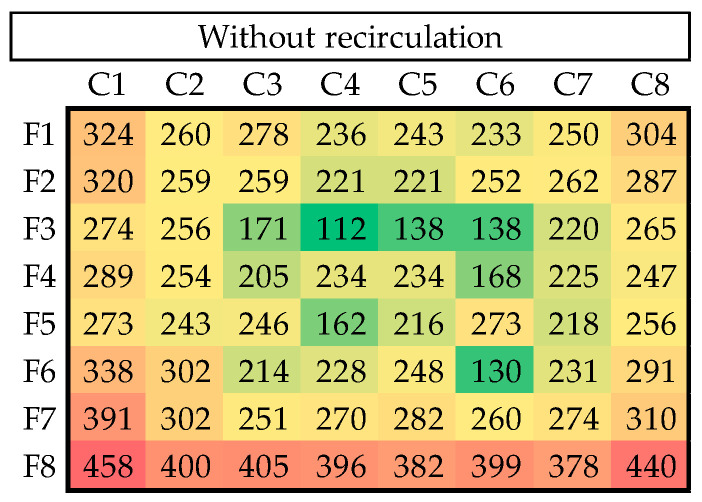
Thickness distribution for test without recirculation.

**Figure 19 materials-18-01972-f019:**
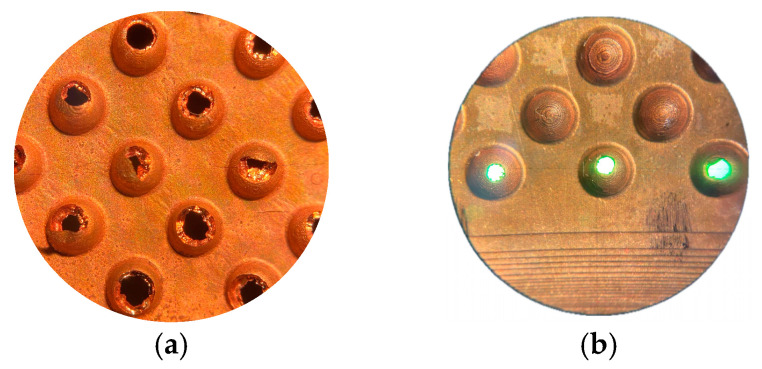
Copper shell defects: (**a**) incomplete filling of a large number of cavities; (**b**) incomplete filling of the lower part of the functional zone of the electrode.

**Figure 20 materials-18-01972-f020:**
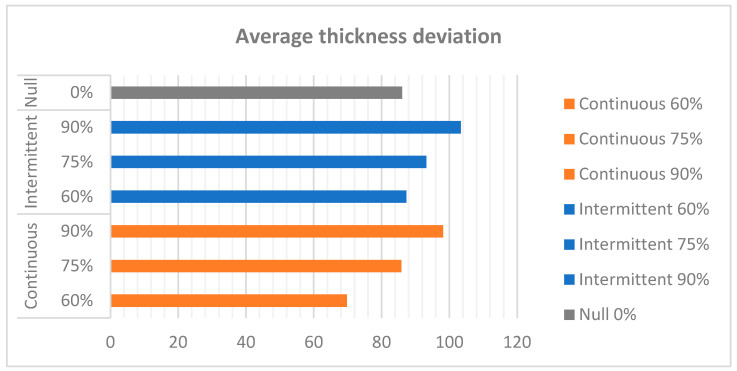
Average shell thickness deviation.

**Figure 21 materials-18-01972-f021:**
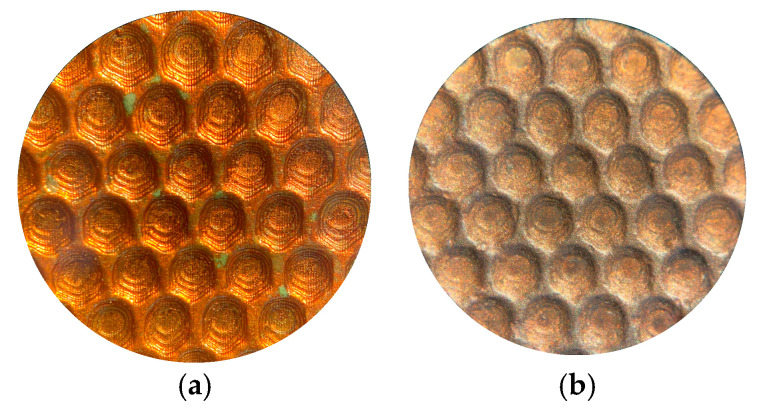
Sharkskin texture on relief: (**a**) electroformed shell; (**b**) eroded shell.

**Table 1 materials-18-01972-t001:** Deposition program used.

Stages	Intensity	Time
1	100 mA	1 h
2	250 mA	30 min
3	500 mA	1 h
4	750 mA	1 h
5	1 A	1 h

**Table 2 materials-18-01972-t002:** Experiment design to analyze the influence of bath agitation.

Experiment	Repetition	Pump Power	Recirculation Type
1	3	60%	Continuous
2	3	75%	Continuous
3	3	90%	Continuous
4	3	60%	Intermittent
5	3	75%	Intermittent
6	3	90%	Intermittent
7	3	0%	Null

## Data Availability

The original contributions presented in this study are included in the article. Further inquiries can be directed to the corresponding author.

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
