# Peer review of "Improved Design of Electroforming Equipment for the Manufacture of Sinker Electrical Discharge Machining Electrodes with Microtextured Surfaces"

_materials, 2025, doi:10.3390/ma18091972_

Round 1
Reviewer 1 Report
Comments and Suggestions for Authors
- Regarding the importance of controlled flow for achieving uniform layer thickness:
The authors are kindly requested to provide additional quantitative data or a comparative analysis to further illustrate the extent of improvement in layer uniformity achieved through the various flow strategies tested.
- Concerning intermittent recirculation and moderate pumping power:
Can the authors elaborate on the specific parameters used for "moderate" pumping power and the timing of intermittent recirculation? How were these optimized, and are they generally applicable to other materials or systems? - In addition to that, authors mention that excessive recirculation leads to turbulence, which negatively impacts deposition uniformity. Could the authors clarify how turbulence was characterized or measured in their study, and provide insight into the threshold at which recirculation becomes detrimental?
It is kindly suggested that the authors consider revising the abstract and conclusion to more clearly and quantitatively highlight the main findings of the study.
Author Response
Dear Reviewer,
We would like to express our sincere gratitude for the time and dedication you have invested in reviewing our manuscript. Your comments have contributed significantly to enriching and strengthening the quality and clarity of our work. We have meticulously reviewed each comment and suggestion, and we have made the necessary modifications to the text. These modifications were made with the aim of improving the presentation, methodological soundness, and interpretation of the results.
We have taken the liberty of addressing each of the points raised below.
Comments 1: Regarding the importance of controlled flow for achieving uniform layer thickness: The authors are kindly requested to provide additional quantitative data or a comparative analysis to further illustrate the extent of improvement in layer uniformity achieved through the various flow strategies tested. |
Response 1: We appreciate this observation. We have expanded the Results section (section 3.1) by incorporating an additional quantitative analysis on the standard deviation of the thickness measured in one of the tests. Specifically, the mean deviation values in microns for each flow configuration were added. For instance, the experiments involving continuous recirculation at 60% power exhibited an average deviation of 70 µm, while the absence of recirculation registered values around 86 µm. This quantitative analysis is reflected in Figures 15 to 18, which now include color maps, as well as the improved graph in Figure 20, which reinforces the understanding of the improvements achieved. |
Comments 2: Concerning intermittent recirculation and moderate pumping power: Can the authors elaborate on the specific parameters used for "moderate" pumping power and the timing of intermittent recirculation? How were these optimized, and are they generally applicable to other materials or systems? |
Response 2: We appreciate this suggestion, as it provides much-needed clarity. As outlined in section 2.3.1, the precise parameters of power types and their associated flow rates have been specified. Regarding intermittent recirculation, a cycle of 10 minutes of activation followed by 5 minutes of rest was programmed using a programmable digital timer, which is also outlined in this section. These values were selected after a series of preliminary tests (not included in this article for space reasons), observing the balance between uniform deposition and absence of surface defects due to swirls or accumulations. While this strategy is optimized for copper and the equipment described, we believe that the sequential control strategy can be adapted to other systems, provided they are rescaled according to their characteristics. Comments 3: In addition to that, authors mention that excessive recirculation leads to turbulence, which negatively impacts deposition uniformity. Could the authors clarify how turbulence was characterized or measured in their study, and provide insight into the threshold at which recirculation becomes detrimental? |
Response 3: Thank you for this critical observation. While turbulence was not directly measured in this study, qualitative characterization was performed through two indicators: (1) the presence of irregular patterns in the deposition, as observed under optical inspection and in thickness measurements, and (2) incomplete filling of cavities in low-relief geometries. From these data, we determined that powers above 75% of the pump generated consistent negative effects in all repetitions, with 90% being clearly detrimental. Consequently, this threshold has been established as the upper operating limit within our system. This criterion has been incorporated into the Discussion as an empirical observation that can be verified in future work using flow visualization techniques. Comments 4: It is kindly suggested that the authors consider revising the abstract and conclusion to more clearly and quantitatively highlight the main findings of the study. |
Response 4: We are taking this suggestion under advisement. The summary and conclusions have been revised to include quantifiable data. For instance, it is now emphasized that the redesign of the equipment enabled a thickness deviation reduction of up to 16.8% compared to conditions without recirculation, and that an average thickness of 90-100 µm was attained. The conclusion has also been reinforced by pointing out that these results are especially relevant for applications where high dimensional accuracy is required in SEDM electrodes. |
Reviewer 2 Report
Comments and Suggestions for Authors
This study presents an innovative methodology combining mask stereolithography and optimized electroforming to fabricate high-precision copper shells for SEDM electrodes, demonstrating enhanced deposit uniformity and quality with low turbulent flows and continuous recirculation, thus providing a viable and efficient alternative for producing textured electrodes with high geometric fidelity.
1.One of the keywords is "additive manufacturing", but I did not see any detailed introduction in the abstract and introduction sections. Please choose a more specific field to use as a keyword. In addition, using "surfaces" as a standalone keyword is a bit too simplistic; please modify it by adding a qualifier in front.
2.Why is the term "redesign" used in the title? Which original design is it being compared to? This is not clearly explained here. To help readers understand better, please modify it to phrases like "design optimization method."
3.In the summary, you mentioned that low turbulent flows perform the best, but in the conclusion section, you only discussed intermittent recirculation and did not mention low turbulent flows. Do the two represent the same category?
4.In Figure 10, I noticed that the two corresponding images have the same scale, but the content of the images is different. Why is there no specific description of the subfigures in Figure 10 in the relative text?
5.In the second part, there is only one subheading 2.1, and the subsequent 2.2 and 2.3 do not exist. Therefore, is it necessary to promote the third-level headings to second-level headings to ensure the rationality of the structure?
Author Response
Dear Reviewer,
We would like to express our sincere gratitude for the time and dedication you have invested in reviewing our manuscript. Your comments have contributed significantly to enriching and strengthening the quality and clarity of our work. We have meticulously reviewed each comment and suggestion, and we have made the necessary modifications to the text. These modifications were made with the aim of improving the presentation, methodological soundness, and interpretation of the results.
We have taken the liberty of addressing each of the points raised below:
Comments 1: One of the keywords is "additive manufacturing", but I did not see any detailed introduction in the abstract and introduction sections. Please choose a more specific field to use as a keyword. In addition, using "surfaces" as a standalone keyword is a bit too simplistic; please modify it by adding a qualifier in front. |
Response 1: We appreciate this comment. We have revised and updated both the keywords and the content of the abstract and introduction. · Specifically, the term "additive manufacturing" has been substituted with the more precise term "mask stereolithography (MSLA)," which more accurately reflects the technique employed. · Additionally, the term "surfaces" has been substituted with "microtextured surfaces" to enhance thematic clarity. |
Comments 2: Why is the term "redesign" used in the title? Which original design is it being compared to? This is not clearly explained here. To help readers understand better, please modify it to phrases like "design optimization method." |
Response 2: The term "redesign" refers to a previous experimental model used in prior works by the research group. This previous model presented limitations in the recirculation of the electrolyte and the control of operating parameters. This new version incorporates the following: These include: - Bath compartmentalization - Controlled recirculation - Automated and modular system agitation To avoid any potential confusion, the title of the article has been updated to: "Improved design of an electroforming equipment for the fabrication of SEDM electrodes with microtextured surfaces". Comments 3: In the summary, you mentioned that low turbulent flows perform the best, but in the conclusion section, you only discussed intermittent recirculation and did not mention low turbulent flows. Do the two represent the same category? |
Response 3: Thank you for sharing this observation. In the original version, the concepts of turbulent flow and recirculation mode were not correctly differentiated. In the updated manuscript, each type of recirculation and the flow rate relevant to each has been clearly explained. In the conclusions, it has been clarified that excessive recirculation (at 90%) caused turbulence that deteriorated uniformity, while continuous recirculation at 60% achieved the best result. This distinction between flow type and operating mode is crucial to avoid conceptual confusion. Comments 4: In Figure 10, I noticed that the two corresponding images have the same scale, but the content of the images is different. Why is there no specific description of the subfigures in Figure 10 in the relative text? |
Response 4: You are correct, and we appreciate your highlighting this issue. The image has been updated to a simpler version, with only 20x magnification being used, as indicated in the text. Each texture studied is explained. Comments 5: In the second part, there is only one subheading 2.1, and the subsequent 2.2 and 2.3 do not exist. Therefore, is it necessary to promote the third-level headings to second-level headings to ensure the rationality of the structure? Response 5: Thank you for pointing out this inconsistency. The entire hierarchical structure of the manuscript has been revised and in this second part, section 2.1 has been removed, promoting the headings from level 3 to level 2 to ensure clear logic. The sections are now organized as: - 2.1 Design and fabrication of functional models. - 2.2 Sputtering - 2.3 Electroforming - 2.4 Use of shells as tooling - 2.5 Validation method This modification improves the clarity of the manuscript and its editorial consistency. |
Round 2
Reviewer 2 Report
Comments and Suggestions for Authors
The authors have revised the manuscript according to the comments. This paper can be accepted.